# Impact of Tree Species Mixture on Microbial Diversity and Community Structure in Soil Aggregates of *Castanopsis hystrix* Plantations

**DOI:** 10.3390/microorganisms13030578

**Published:** 2025-03-03

**Authors:** Rongao Zhang, Yijun Liu, Fei Cheng

**Affiliations:** 1Guangxi Colleges and Universities Key Laboratory for Cultivation and Utilization of Subtropical Forest Plantation, College of Forestry, Guangxi University, Nanning 530004, China; f2512936835@163.com (R.Z.); liu_yijun2021@163.com (Y.L.); 2Guangxi Key Laboratory of Forest Ecology and Conservation, College of Forestry, Guangxi University, Nanning 530004, China; 3Department of Forestry, College of Forestry, Guangxi University, Nanning 530004, China

**Keywords:** *Castanopsis hystrix*, mixed forests, soil aggregates, microbial community

## Abstract

Soil aggregates play a crucial role in maintaining the health and stability of artificial forest soil ecosystems, and microorganisms contribute to the formation and maintenance of soil aggregates. However, the impact of different tree species in mixed forests on soil aggregate microbial communities remains unclear. In this study, high-throughput sequencing technology was employed to analyze the bacterial and fungal diversity and community composition of four soil aggregate sizes (<0.25 mm, 0.25–1 mm, 1–2 mm, and >2 mm) in pure *Castanopsis hystrix* plantations (CK), mixed *C. hystrix* and *Acacia crassicarpa* plantations (MCA), mixed *C. hystrix* and *Pinus massoniana* plantations (MCP), and mixed *C. hystrix* and *Mytilaria laosensis* plantations (MCM). The results indicate that (1) establishing mixed forests enhances the diversity of bacterial and fungal communities in soil aggregates, and that soil aggregates with size <0.25 mm support higher microbial diversity. (2) The fungal and bacterial composition of soil aggregates in mixed forests differs from that of pure *C. hystrix* forests. The dominant bacterial phyla in the four forest types are Proteobacteria, Acidobacteria, Actinobacteria, and Chloroflexi. The dominant fungal phyla are Basidiomycota, Ascomycota, Mortierellomycota, and Mucoromycota. (3) PCoA analysis reveals that compared to pure *C. hystrix* forests, mixing with *A. crassicarpa* (MCA) results in marked changes in the bacterial community structure of soil aggregates; similarly, mixing with *A. crassicarpa* (MCA) and *M. laosensis* (MCM) leads to significant differences in the fungal community structure of soil aggregates. (4) RDA results show that NH_4_^+^-N, pH, and OC are the main factors influencing microbial diversity in soil aggregates. In terms of dominant microorganisms, pH and AP are the key environmental factors affecting the structure of bacterial and fungal communities in soil aggregates. The findings of this study contribute to our understanding of the characteristics of microbial communities in soil aggregates affected by tree mixing and provide a scientific reference for the maintenance and enhancement of soil fertility in planted forests.

## 1. Introduction

Plantation forests, as a crucial component of terrestrial ecosystems, play a significant role in the supply of forest products, the enhancement of carbon sequestration, and the improvement of ecological environments [1]. To maintain and enhance the productivity and ecological benefits of plantation forests, establishing mixed forests with complex stand structures and high biodiversity has become an important direction in current forestry development [2]. Compared to pure forests, mixed forests exhibit obvious advantages in terms of biomass, structural stability, and soil properties, such as reduced soil bulk density, regulated pH, and increased contents of organic matter, NPK, and trace elements.

Soil aggregates, as structural units of soil, can undergo changes in their composition when mixed with tree species that produce abundant litter and developed root systems [3]. The relative proportions of soil aggregates of different sizes in mixed forests differ significantly from those in pure forests. Mixing tree species can significantly increase the proportion of large-sized aggregates [4], although some studies have reported the opposite [5]. Mixing can also effectively enhance the stability of soil aggregates, which may be related to the rate of organic matter transformation [6], increased storage of organic carbon and other nutrient elements [7], and increased ectomycorrhizal fungi [8]. Some studies have shown that soil aggregates in mixed forests are richer in organic matter, nitrogen, phosphorus, and available nutrients, and their distribution is closely related to aggregate size and relative proportion [9]. However, little is known about how mixing affects the distribution of soil aggregate-associated microorganisms and the effects of different tree species.

Soil aggregates of different sizes differ in distribution, stability, physicochemical properties, and pore characteristics [10,11,12], which directly influence the distribution of water and air within the soil and the material and energy exchange between microorganisms and the external environment [13]. As a special type of forest, mixed forests have significant impacts on soil aggregates and their microbial communities due to the diversity and high coverage of litter [14,15,16]. Mixed forests reduce the destruction of soil aggregates by runoff and rainwater [17], promote the binding and aggregation of aggregates [8,18], and thereby indirectly affect microbial activity, diversity, and community structure within soil aggregates. Specifically, the introduction of broad-leaved species in mixed forests can increase microbial biomass and enzyme activity in soil aggregates and affect the rate of organic matter decomposition [19], which is considered to be the result of increased litter input providing more C and N sources for microorganisms. Changes in tree species diversity also affect the efficiency of microbial communities in utilizing C [20]. Introducing arborous broad-leaved species into pure *Cunninghamia lanceolata* (Pinopsida: Cupressaceae) forests to form multi-layered mixed forests can significantly increase the storage of organic carbon and nutrients in soil aggregates [18], providing a more favorable environment for microbial development. Different mixed tree species not only alter the composition and structure of the stand but also stimulate the production of aggregate-associated microbial derivatives through their root exudates and litter. The polymers, mucilage, and other organic acid compounds secreted by microorganisms can promote soil aggregation [21] while enriching the types of organic matter and providing abundant resources for the coexistence of diverse microorganisms. Changes in the physicochemical properties of soil aggregates caused by mixing can also simultaneously affect the microbial community structure [22,23,24,25,26]. Despite some progress in research on soil microorganisms in mixed forests, most studies have focused on the entire soil level without distinguishing between aggregates. Little is still known about how mixing affects the distribution of soil aggregate-associated microorganisms and the effects of different tree species.

*Castanopsis hystrix*, with its fast growth, excellent wood quality, and strong adaptability [27], has become one of the important broad-leaved timber species in southern China. However, the traditional mode of pure *C. hystrix* forests has caused a series of ecological problems, including reduced biodiversity, soil fertility decline, and significant productivity reduction [28]. Establishing mixed forests of *C. hystrix* has gradually emerged as one of the most promising forest management approaches to address these issues. Currently, research on soil aggregates in mixed forests of *C. hystrix* mainly focuses on the early impacts on soil aggregate organic carbon and its components [16], the effects on soil aggregate phosphorus fractions and their transformation [29], and the effects on soil aggregate distribution and stability [30]. However, studies on the microbial diversity and community structure of soil aggregates have not been reported. Therefore, this study takes pure *C. hystrix* forests (CK), mixed forests of *C. hystrix* with *Acacia crassicarpa* (Magnoliopsida: Fabaceae) (MCA), mixed forests of *C. hystrix* with *Pinus massoniana* (Pinopsida: Pinaceae) (MCP), and mixed forests of *C. hystrix* with *Mytilaria laosensis* (Magnoliopsida: Hamamelidaceae) (MCM) as research objects to elucidate the diversity and community structure characteristics of microorganisms in soil aggregates of different sizes and explore the key limiting factors affecting soil aggregate-associated microbial diversity and community structure. This will provide scientific references for the transformation of pure *C. hystrix* forests into mixed forests and soil fertility maintenance. The hypotheses of this study are (1) compared to pure *C. hystrix* forests, mixed forests of *C. hystrix* increased the nutrient level of soil aggregates and thus changed the microbial community structure and diversity of aggregates; (2) under the same stand, the nutrient levels of soil aggregates with different grain grades are different, and small aggregates support higher nutrient levels and thus show higher microbial diversity.

## 2. Materials and Methods

### 2.1. Study Area

The study area is located within the Gaofeng Forest Park, Guangxi Autonomous Region, China (22°49′~23°15′N, 108°8′~108°53′E). The park’s landform primarily consists of hills and mountains, with elevations ranging from 150 to 400 masl and slopes between 5 and 30 degrees. The climate is classified as a south subtropical monsoon climate, characterized by an annual average temperature of approximately 21.5 °C, and an annual average precipitation of approximately 1300 mm. The soil type is mostly lateritic red soil, which is acidic or slightly acidic. The soil parent material is mainly sandstone and mudstone, and over 80% of the park has medium to thick soil layers with good water and fertilizer retention.

### 2.2. Forest Stands

Based on field investigations conducted in September 2023, four forest stands with the same soil parent material and similar slope aspects and gradients were selected for study: pure *C. hystrix* forest (CK), *C. hystrix*-*A. crassicarpa* mixed forest (MCA), *C. hystrix*-*M. laosensis* mixed forest (MCM), and *C. hystrix*-*P. massoniana* mixed forest (MCP). All four forest stands were planted in 2002 with a spacing of 3 × 4 m. The mixed forests have a mixing ratio of 1:1 and are arranged in an alternating row pattern. Soil loosening, mechanical weeding without herbicides, and the application of compound fertilizer (N:P:K = 15:15:15) at 60 kg per acre were carried out annually between 2003 and 2006. Since then, there have been no significant natural disasters or human disturbances. Slope measurements are taken on a slope scale, and slope corrections are required for steeper slopes. We also carried out the measurement of forest floor cover via canopy projection, Broules altimeter measures of tree height, and diameter tape measures of the diameter at chest height. The environmental and tree growth information for the four forest stands is presented in Table 1.

### 2.3. Plot Establishment and Sample Collection

Three 20 × 20 m plots were established in each forest stand, with a distance of more than 25 m between replicate plots, totaling 12 plots. Within each plot, five representative sampling points were selected along an “S”-shaped path at the edge of the tree canopy, away from the root system. After removing the surface litter, intact soil samples from the 0–10 cm soil layer were collected using PVC tubes (20 cm high with a 10 cm inner diameter), and soil samples were also collected using a soil core sampler with a volume of 100 cm^3^. The intact soil samples from each plot were mixed to obtain 12 composite soil samples, which were then placed in sterile boxes, labeled, and stored in foam boxes with dry ice for transportation back to the laboratory. Using the best possible aseptic conditions, such as a UV-sterilized workbench, sterilized gloves, and sterilized sieves, the soil samples were pretreated by carefully separating them along their natural textures to protect the aggregate structure from damage and sieving out small stones and plant and animal residues using a 5 mm sieve. The pretreated soil samples were divided into two portions: one for determining the chemical characteristics of the whole soil and the other for sieve analysis when the soil moisture content was approximately 10%. A 500 g aliquot of the pretreated soil sample was sieved for 10 min using a standard vibrating sieve machine with aperture sizes of 2, 1, and 0.25 mm to obtain four aggregate size fractions: <0.25 mm, 0.25–1 mm, 1–2 mm, and >2 mm. The weights of the aggregates in each size fraction were measured. The soil aggregate samples were sent to Beijing Novogene Company for microbial sequencing amplicon, with dry ice used for preservation during transportation.

### 2.4. Determination of Abiotic Characteristics of Bulk Soil and Aggregates

#### 2.4.1. Aggregate Size Distribution and Stability Parameter

The composition of aggregate distribution was obtained by weighing the mass of four aggregate sizes and calculating the proportion of each size. The mean weight diameter (MWD) was calculated using the following formula:(1)$$MWD=\sum_{i=1}^{n}\left({\bar{x}}_{i\;}w_{i}\right)$$

In the formula, ${\bar{x}}_{i}$ denotes the average diameter of aggregates within each size, and $w_{i}$ denotes the mass percentage of aggregates within each size. The aggregate size distribution and stability are presented in Table 2.

#### 2.4.2. Physicochemical Factors

Bulk density (BD) and soil porosity (SP) of the whole soil were determined using the core method. The pH of the whole soil and aggregates was measured using the potentiometric method. The water content (WC) of the whole soil was determined by oven drying. Organic carbon (OC) content in the whole soil and aggregates was assayed using the potassium dichromate oxidation-external heating method. Total nitrogen (TN) content was analyzed using the Kjeldahl nitrogen determination method. Total phosphorus (TP) content was determined using the Kjeldahl digestion-molybdenum antimony anti-colorimetric method, while available phosphorus (AP) content was measured using the double acid extraction-molybdenum antimony anti-colorimetric method. Ammonium nitrogen (AN) and nitrate nitrogen (NN) contents in the whole soil and aggregates were assayed using potassium chloride extraction followed by flow injection analysis. The physicochemical properties of the whole soil and aggregates are presented in Table 2 and Table 3, respectively.

**Table 2 microorganisms-13-00578-t002:** Physico-chemical properties of soil aggregates in different stand types.

Indicator	Stand Type	Soil Aggregate Size			
		>2 mm	1–2 mm	0.25–1 mm	<0.25 mm
pH	CK	4.51 ± 0.02 Aa	4.50 ± 0.00 Aa	4.50 ± 0.02 Aa	4.50 ± 0.05 Aa
	MCA	4.15 ± 0.02 Aa	4.18 ± 0.05 Ab	4.23 ± 0.11 Ab	4.24 ± 0.01 Ab
	MCM	4.52 ± 0.03 Aa	4.47 ± 0.07 Aa	4.47 ± 0.11 Aa	4.54 ± 0.08 Aa
	MCP	4.50 ± 0.12 Aa	4.47 ± 0.10 Aa	4.46 ± 0.13 Aa	4.48 ± 0.10 Aa
OC (g/kg)	CK	17.4 ± 2.72 Bb	21.1 ± 2.11 ABa	23.9 ± 1.76 Ab	22.2 ± 0.72 Ab
	MCA	28.8 ± 0.88 Ba	26.9 ± 1.82 Ba	32.3 ± 1.54 Aa	33.3 ± 1.86 Aa
	MCM	18.7 ± 2.45 Ab	20.1 ± 2.59 Aa	22.9 ± 3.99 Ab	24.9 ± 1.36 Ab
	MCP	20.6 ± 2.16 Ab	22.4 ± 4.78 Aa	23.6 ± 4.02 Ab	24.8 ± 3.95 Ab
TN (g/kg)	CK	1.35 ± 0.17 Ca	1.62 ± 0.12 Ba	1.85 ± 0.05 Aa	1.75 ± 0.07 ABa
	MCA	1.83 ± 0.30 Aa	1.78 ± 0.27 Aa	2.11 ± 0.46 Aa	2.14 ± 0.51 Aa
	MCM	1.38 ± 0.11 Ba	1.54 ± 0.15 Ba	1.73 ± 0.26 ABa	1.90 ± 0.17 Aa
	MCP	1.51 ± 0.17 Aa	1.53 ± 0.22 Aa	1.65 ± 0.23 Aa	1.70 ± 0.21 Aa
TP (g/kg)	CK	0.24 ± 0.03 Aa	0.23 ± 0.01 Aa	0.23 ± 0.01 Aa	0.22 ± 0.01 Aa
	MCA	0.22 ± 0.02 Aa	0.21 ± 0.02 Aa	0.22 ± 0.03 Aa	0.22 ± 0.04 Aa
	MCM	0.23 ± 0.01 Aa	0.22 ± 0.02 Aa	0.24 ± 0.01 Aa	0.23 ± 0.02 Aa
	MCP	0.22 ± 0.04 Aa	0.21 ± 0.02 Aa	0.23 ± 0.05 Aa	0.23 ± 0.05 Aa
NH_4_^+^-N (mg/kg)	CK	35.4 ± 2.09 Ba	46.0 ± 3.95 ABa	53.9 ± 6.38 Aa	51.8 ± 11.15 Ab
	MCA	24.3 ± 6.40 Ab	25.7 ± 6.61 Ab	33.3 ± 9.20 Ab	32.0 ± 7.06 Ac
	MCM	40.7 ± 5.70 Ca	43.5 ± 0.79 Ca	55.8 ± 2.44 Ba	66.5 ± 5.24 Aa
	MCP	37.6 ± 2.93 Ca	42.3 ± 1.49 BCa	45.0 ± 3.56 ABa	48.7 ± 3.09 Ab
NO_3_^−^-N (mg/kg)	CK	1.62 ± 0.14 Aa	1.27 ± 0.15 Ac	1.50 ± 0.14 Ac	1.75 ± 0.46 Ac
	MCA	3.96 ± 0.51 Ab	3.71 ± 0.48 Ab	3.91 ± 0.70 Ab	3.55 ± 0.58 Ab
	MCM	3.00 ± 0.58 Bc	6.44 ± 1.35 Aa	6.29 ± 1.21 Aa	5.56 ± 0.54 Aa
	MCP	5.96 ± 0.13 Ba	6.58 ± 0.25 Aa	6.48 ± 0.13 ABa	6.43 ± 0.46 ABa
AP (mg/kg)	CK	5.65 ± 0.49 Bab	6.00 ± 0.22 Bb	7.10 ± 0.40 Aa	7.82 ± 0.75 Aa
	MCA	6.50 ± 0.76 Aa	6.40 ± 1.08 Ab	7.53 ± 1.45 Aa	7.05 ± 1.65 Aa
	MCM	3.58 ± 0.23 Bc	3.13 ± 0.52 Bc	4.15 ± 0.87 Bb	6.13 ± 0.90 Aa
	MCP	5.33 ± 0.23 Cb	11.3 ± 1.62 Aa	6.40 ± 0.45B Ca	7.28 ± 0.28 Ba
C/N	CK	12.9 ± 0.46 Aa	13.1 ± 0.52 Aa	12.9 ± 0.61 Aa	12.7 ± 0.20 Aa
	MCA	16.0 ± 2.33 Aa	15.2 ± 1.25 Aa	15.7 ± 3.18 Aa	16.1 ± 3.41 Aa
	MCM	13.5 ± 1.31 Aa	13.0 ± 0.81 Aa	13.2 ± 0.31 Aa	13.1 ± 0.46 Aa
	MCP	13.6 ± 0.45 Aa	14.5 ± 1.17 Aa	14.3 ± 0.75 Aa	14.5 ± 0.58 Aa
C/P	CK	75.1 ± 16.9 Bb	91.3 ± 11.5 ABa	106 ± 5.60 Ab	102 ± 4.80 Ab
	MCA	133 ± 5.64 Aa	130 ± 8.21 Aa	147 ± 17.1 Aa	155 ± 19.8 Aa
	MCM	82.0 ± 14.2 Ab	91.4 ± 17.5 Aa	97.4 ± 18.4 Ab	109 ± 1.50 Ab
	MCP	96.0 ± 18.4 Ab	107 ± 27.6 Aa	106 ± 24.9 Ab	109 ± 18.6 Ab
N/P	CK	5.79 ± 1.21 Ba	6.98 ± 0.71 ABa	8.26 ± 0.37 Aa	8.07 ± 0.27 Ab
	MCA	8.47 ± 0.82 Aa	8.58 ± 0.71 Aa	9.44 ± 0.78 Aa	9.72 ± 0.90 Aa
	MCM	6.05 ± 0.83 Aa	7.00 ± 1.11 Aa	7.38 ± 1.22 Aa	8.30 ± 0.22 Aab
	MCP	7.08 ± 1.61 Aa	7.37 ± 1.47 Aa	7.45 ± 1.85 Aa	7.54 ± 1.25 Ab
Composition (%)	CK	28.0 ± 0.16 b	27.6 ± 0.02 a	33.8 ± 0.01 ab	10.4 ± 0.01 b
	MCA	38.8 ± 0.02 a	26.7 ± 0.06 c	25.0 ± 0.02 a	9.46 ± 0.00 b
	MCM	33.1 ± 0.01 ab	32.6 ± 0.01 b	27.4 ± 0.01 b	6.93 ± 0.00 ab
	MCP	34.7 ± 0.06 a	30.5 ± 0.03 a	28.1 ± 0.04 b	6.59 ± 0.01 a
MWD (mm)	CK	1.62 ± 0.03 b			
	MCA	1.93 ± 0.05 a			
	MCM	1.83 ± 0.03 a			
	MCP	1.86 ± 0.14 a			

Note: CK, pure *C. hystrix* plantations; MCA, mixed *C. hystrix* and *A. crassicarpa* plantations; MCM, mixed *C. hystrix* and *M. laosensis* plantations; MCP, mixed *C. hystrix* and *P. massoniana* plantations. OC, organic carbon; TN, total nitrogen; TP, total phosphorus; AP, available phosphorus; NH_4_^+^-N, ammonium nitrogen; NO_3_^–^-N, nitrate nitrogen; C/N, carbon to nitrogen ratio; C/P, carbon to phosphorus ratio; N/P, nitrogen to phosphorus ratio; MWD, mean weight diameter. Capital letters indicate differences in aggregate sizes of same stand type at the 0.05 significant level, lowercase letters indicate differences in stand types of same aggregate size at the 0.05 significant level.

### 2.5. High-Throughput Sequencing of Soil Aggregate Microbiota

A 0.25 g aliquot of soil aggregate samples was used to extract microbial DNA using the Solarbio Soil DNA Kit (D2600, Solarbio Science, Beijing, China). Following DNA extraction, PCR amplification was performed using primers targeting the bacterial 16S V4 region (515F and 806R) and the fungal ITS 1 region (1737F and 2043R) [31,32]. The PCR reaction protocol included an initial denaturation at 98 °C for 1 min, followed by 30 cycles of denaturation at 98 °C for 10 s, annealing at 50 °C for 30 s, and extension at 72 °C for 30 s, with a final extension at 72 °C for 5 min. PCR products were then quantified, normalized, and mixed, followed by agarose gel electrophoresis (2.0%) to detect and recover the target bands. Libraries were constructed, quantified using Qubit and Q-PCR, and evaluated for quality before sequencing on a NovaSeq (6000, Illumina, San Diego, CA, USA) with PE250, performed by Beijing Novogene Company.

After obtaining raw sequencing data, preprocessing was conducted to ensure the reliability of the information analysis, including three steps: ① quality filtering: raw sequences were first screened using Trimmomatic (v0.33, Broad Institute, Cambridge, MA, USA), and primers were identified and removed using Cutadapt (1.9.1, TU Dortmund, Dortmund, Germany) to obtain Clean Reads without primer sequences; ② paired-end sequence merging: clean Reads from each sample were merged using overlap, and the merged data were filtered based on length within specified ranges for different regions; ③ chimera removal: chimera sequences were identified and removed using UCHIME (4.2, AmyJet Scientific Inc, Wuhan, China) to obtain valid data; ④ denoising: denoising was performed on valid data using DADA (2, Jargeon, Guangzhou, China) to obtain final ASVs (amplicon sequence variants).

### 2.6. Statistical Analysis

Data were organized using Microsoft Excel 2020, and the homogeneity of variances for microbial abundance and diversity indices among different treatments was tested using Levene’s Test in SPSS (27.0, IBM, Armonk, NY, USA). A one-way ANOVA was then conducted to test differences between different forest types or aggregate sizes. If *p* > 0.05, there was no significant difference between treatments; if *p* < 0.05, Duncan’s multiple range test was further performed. For microbial diversity analysis, representative sequences of each ASV were annotated to obtain corresponding species and abundances. The richness and diversity of bacteria and fungi within relevant samples, as well as the number of shared and unique ASVs between different forest types or aggregate sizes, were analyzed. Origin (2022, OriginLab, Northampton, MA, USA) was used to plot species abundance maps and Venn diagrams for each stand type. Principal coordinates analysis (PCoA) plots were constructed based on distance matrices using abundance information of characteristic sequences. Redundancy analysis (RDA) plots were generated using Canoco (5.0, Biometris, Shanghai, China). The construction of partial least squares models (PLS-PM) was performed using R (4.3.3, UoA, Auckland, New Zealand) and the “Plspm” package (plspm, Gaston Sanchez, Berkeley, CA, USA).

## 3. Results

### 3.1. Physical and Chemical Environment of Forest Land and Soil

Under the same aggregate size, except TP and C/N, stand types significantly affected soil aggregates TN, OC, NH_4_^+^-N, NO_3_^−^-N, AP, pH, C/P, and N/P (*p* < 0.05, Table 2). In the same stand, soil aggregate size significantly affected soil aggregate TN, OC, NH_4_^+^-N, NO_3_^−^-N, AP, C/P, and N/P in addition to TP, PH, and C/N (*p* < 0.05, Table 2). The contents of TN, OC, and NH_4_^+^-N increased overall with the decrease in the size of the soil aggregates (Table 2).

### 3.2. Microbial Diversity in Soil Aggregates

Across all aggregate sizes, the bacterial Chao1 index was lowest in MCA, particularly reaching significant levels in the 1–2 mm and 0.25–1 mm aggregate sizes (Figure 1a). The Chao1 index for these two aggregate sizes in MCP was also lower than that in CK and MCM. The bacterial Shannon index was lowest in MCA for all aggregate sizes (Figure 1b), showing significant differences among the three aggregate sizes smaller than 2 mm, while there were generally no significant differences among the other three forest types.

For fungi, the Chao1 and Shannon indices were highest in MCM for aggregates larger than 2 mm and smaller than 0.25 mm, with generally no significant differences among the other three forest types (Figure 1c,d). Significant differences in these two indices were observed among different forest types for the other two aggregate sizes, but CK and MCM were generally higher, while MCA and MCP were relatively lower.

### 3.3. Analysis of Microbial Community Composition in Soil Aggregates

At the phylum level, Proteobacteria, Acidobacteriota, Actinobacteriota, and Chloroflexi dominated the bacterial communities in soil aggregates across all stand types, with the combined relative abundance of these four phyla ranging from 75.55% to 86.21% (Figure 2a). Differences in bacterial phylum composition were observed among different stand types. In general, Acidobacteriota and Proteobacteria had the highest relative abundances among the four dominant phyla in soil aggregates of various sizes in CK, MCM, and MCP, followed by Actinobacteriota and Chloroflexi. In contrast, Actinobacteriota had a higher relative abundance in MCA across all aggregate sizes, with a corresponding decrease in Acidobacteriota.

At the genus level (Figure 2b), the top 10 bacterial genera accounted for 19.06% to 38.13% of the total relative abundance, with *Acidothermus*, *Acidibacter*, *Solibacter* Candidate, and *Bryobacter* being prominent. Differences in bacterial genus composition were observed among different stand types. Excluding “other,” *Achromobacter* was almost absent in all stands except CK. *Acidothermus* had significantly higher relative abundances in MCA across all aggregate sizes compared to other stands, while *Solibacter* Candidate had lower relative abundances. For CK, MCM, and MCP, the abundances of *Acidothermus* and *Solibacter* Candidate were similar across samples, except for slightly lower abundances in CK’s >2 mm aggregate sizes.

At the phylum level (Figure 2c), Basidiomycota fungi dominated the community in most cases (25.96% to 84.94%), followed by Ascomycota (9.92% to 25.62%), with smaller amounts of Mortierellomycota (0.1% to 16.54%) and Mucoromycota (0.83% to 2.43%). Differences in fungal phylum composition were observed among different stand types. Although the relative abundances of Basidiomycota and Ascomycota fluctuated across different aggregate sizes in other stands, Basidiomycota consistently had a significantly higher relative abundance than Ascomycota. The relative abundance of Basidiomycota fungi in soil aggregates of various sizes in MCM was significantly reduced due to the influence of “other” and Mortierellomycota.

At the genus level (Figure 2d), excluding “other,” all stands had a high abundance of uncertain-positioning genus *Thelephoraceae* fungi. However, due to the influence of “other,” MCM had significantly lower abundances across all aggregate sizes compared to other stands. In addition to the uncertain-positioning genus *Thelephoraceae*, CK had higher proportions of *Russula* and *Clavulina* in aggregates of various sizes, with little *Elaphomyces*. In contrast, MCA not only had significantly higher abundances of uncertain-positioning genus *Thelephoraceae* across all aggregate sizes compared to CK but also had high abundances of *Elaphomyces*, with little *Russula* and *Clavulina*. MCP had higher abundances of the uncertain-positioning genus *Thelephoraceae* in three out of four aggregate sizes, followed by *Clavulina*, while *Russula* and *Elaphomyces* also had certain abundances.

### 3.4. Distribution of ASV Counts in Soil Aggregates

Among the treated samples, the bacterial community had 22,444 ASVs, of which 0.29% were unassigned, and the fungal community had 7844 ASVs, of which 37.06% were unassigned. Tree mixing had a significant effect on increasing the number of unique bacterial ASVs in soil aggregates larger than 2 mm in diameter (Figure 3a). At this aggregate size, the unique ASV counts in the three mixed stands were 1.23 to 1.61 times higher than those in the control stand (CK) (Figure 3). For aggregates of other sizes, except for MCM, which had unique ASV counts comparable to or higher than CK, the unique ASV counts in the other two mixed stands were mostly lower than CK (Figure 3). The promotion effect of mixing on the number of unique fungal ASVs was particularly evident in aggregates of all sizes in MCM, which had unique ASV counts 1.27 to 2.15 times higher than CK (Figure 4). MCA came next, with unique ASV counts comparable to or slightly higher than CK across different aggregate sizes. However, MCP not only failed to increase but actually had a decrease in unique ASV counts compared to CK (Figure 4). At each aggregate size level, the number of bacterial and fungal ASVs shared among all four stands or in pairs/triplets was far lower than the number of unique ASVs in each stand. Overall, tree mixing significantly promoted the number of unique bacterial ASVs in soil aggregates larger than 2 mm in diameter, and the promotion effect on unique fungal ASVs was particularly pronounced in aggregates of all sizes in MCM.

### 3.5. β-Diversity of Microorganisms in Soil Aggregates

The results of PcoA based on ASV revealed that the cumulative explanatory power for the bacterial community in soil aggregates was 63.16% (Figure 5a). Among the stands, MCA exhibited a greater degree of dispersion from the other three stand types, indicating that the bacterial community structure of soil aggregates across different sizes in MCA differed from those in the other stand types. CK, MCM, and MCP clustered together, suggesting that the bacterial community structures of soil aggregates in these three stand types were relatively similar, which implies that the intercropping of *M. lanceolata* and *P. massoniana* with *C. hystrix* had minimal effects on altering the bacterial community structure of soil aggregates. 

For the fungal community in soil aggregates (Figure 5b), the cumulative explanatory power was 49.87%. Specifically, the fungal structures of soil aggregates in CK and MCP were relatively similar, whereas MCA and MCM were distributed in the upper and lower halves of the plot, respectively, and most of them were distant from CK. This suggests that the fungal community structures of soil aggregates in these two mixed forests differed from those in CK. Therefore, intercropping with *A. melanoxylon* led to changes in the bacterial community structure of soil aggregates, while intercropping with *A. melanoxylon* and *M. lanceolata* resulted in differences in the fungal community structure of soil aggregates compared to the pure *C. hystrix* stand. 

### 3.6. Relationships Between Soil Aggregate Chemical Factors and Microbial Communities

#### 3.6.1. Microbial Diversity

The RDA conducted on microbial diversity and soil aggregate chemical factors revealed that the first and second axes collectively explained 92.5% of the variation in bacterial diversity within soil aggregates (Figure 6a). Among the factors, NH_4_^+^-N had the greatest impact on bacterial diversity (*p* = 0.002), followed by OC (*p* < 0.05). Specifically, NH_4_^+^-N showed positive correlations with Chao1 and Shannon indices, while OC exhibited negative correlations with both. Therefore, NH_4_^+^-N and OC in soil aggregates were identified as the primary factors influencing bacterial community structure.

According to Figure 6b, the contribution rates of the first and second axes were 90.92% and 0.05%, respectively, collectively explaining 90.97% of the variance in fungal diversity. NH_4_^+^-N had the most significant impact on fungal diversity (*p* = 0.018), while pH and C/P also had notable effects (*p* < 0.05). NH_4_^+^-N, pH, and C/P showed positive correlations with both Chao1 and Shannon indices of fungal diversity. These results indicate that NH_4_^+^-N, pH, and C/P in soil aggregates are the main factors influencing fungal diversity.

#### 3.6.2. Distribution of Dominant Microbial Taxa

To elucidate the relationship between microbial communities and soil aggregate chemical factors, RDA was conducted on the dominant microorganisms (top four phyla and genera) and chemical factors within soil aggregates of various stand types. The RDA results revealed that the first axis (79.59%) and the second axis (9.3%) collectively explained 88.89% of the variation in the bacterial community structure of soil aggregates (Figure 7a). Soil aggregate pH had the greatest impact on the bacterial community structure (*p* = 0.002), followed by C/N (*p* < 0.05). pH showed positive correlations with Chloroflexi, *Bryobacter*, Acidobacteriota, and *Solibacter* Candidate, while it was negatively correlated with *Acidothermus* and Actinobacteriota. In contrast, C/N exhibited opposite correlations. Therefore, soil aggregate pH and C/N were identified as the primary factors influencing the bacterial community structure.

Figure 7b shows that the first and second axes contributed 61.23% and 20.87% of the variance, respectively, collectively explaining 82.1% of the variance. The fungal community structure of soil aggregates exhibited significant correlations with AP, pH, NH_4_^+^-N, and TN (*p* < 0.05), with AP having the greatest impact (*p* = 0.004). AP was positively correlated with Basidiomycota and negatively with Mortierellomycota. pH and NH_4_^+^-N were negatively correlated with the uncertain-positioning genus *Thelephoraceae*, *Elaphomyces*, and Ascomycota, whereas TN showed positive correlations with these phyla but negative correlations with *Russula*, *Clavulina*, and Mucoromycota. Therefore, soil aggregate AP, pH, NH_4_^+^-N, and TN were identified as the main factors influencing the fungal community structure. 

#### 3.6.3. PLS-PM Analysis

The PLS-PM provided a good fit to the data (GOF = 0.610) and explained 78.7% of the variance in microbial factors (Figure 8). Stand factors exhibited a positive but non-significant effect on the chemical characteristics and stability of soil aggregates. The chemical characteristics of soil aggregates had a significant positive effect on aggregate stability (*p* < 0.05). However, aggregate stability had little impact on microbial factors, whereas the chemical characteristics of soil aggregates exhibited a highly significant negative effect on microbial factors (*p* < 0.01). This further confirms the crucial role played by the chemical characteristics of soil aggregates in shaping microbial diversity and community structure within soil aggregates in mixed forests.

## 4. Discussion

### 4.1. Relationships Between Soil Aggregate Factors and Microbial Communities

Under similar site conditions, variations in soil microbial community structure are primarily attributed to changes in vegetation type [33]. The results of this study indicate that mixed-tree species planting significantly promotes the number of unique ASVs of bacteria in soil aggregates larger than 2 mm in diameter. The fungal diversity index and richness index in soil aggregates of the MCM are generally higher than those of the other three forest types, suggesting that the establishment of mixed forests can enhance soil microbial diversity, but this effect is dependent on the species of trees mixed. Relevant studies have pointed out that tree species variation is a factor influencing the taxonomic diversity of soil bacterial communities. The composition and availability of litter significantly affect soil nutrient accumulation, thereby altering the diversity of bacteria and fungi in the soil [34,35]. Under different vegetation types, there are notable differences in the nature, quality, quantity, and decomposition rates of surface litter, which have substantial impacts on the input of soil organic components and, consequently, on the structure and diversity of soil microbial communities [36,37]. Typically, the decomposition rate of litter under broadleaf forests is higher than that under coniferous forests [38]. The MCM exhibits richer microbial diversity, possibly because as a broadleaf mixed forest, it has a faster litter decomposition rate, providing more metabolic substrates for microbial proliferation and growth. This study shows that mixing with *A. crassicarpa* generally reduces soil aggregate bacterial diversity. Previous research has found that litter under *A. crassicarpa* contains high lignin content [39]. Rahman et al. [40] suggested that higher lignin concentrations can affect litter decomposition rates. Therefore, the mixed forest with *A. crassicarpa* (MCA) may have reduced its ability to decompose litter due to its relatively high lignin concentration, leading to decreased soil aggregate bacterial diversity. Different aggregate sizes have a significant impact on the diversity of fungal communities. The Chao1 index and Shannon index of fungal communities generally increase as soil aggregate size decreases, indicating that smaller soil aggregates (<0.25 mm) support higher fungal diversity. This finding is similar to the results of Wang et al. [28]. The reason may be that compared to larger soil aggregates, smaller aggregates have smaller pores and slower nutrient turnover, providing diverse habitats and nutrient supplies conducive to microbial growth and reproduction, thereby enhancing microbial diversity. However, Chen [30] found that the diversity index was highest in large aggregates, which is in contrast to the results of this study.

Furthermore, different plant types can input specific secondary metabolites (such as organic acids, sugars, phenols, etc.) into soil aggregates, exerting related positive and negative effects on the species, quantity, and growth and reproduction of aggregate microorganisms [41]. The improvement of soil aggregate stability and appropriate size distribution are important for increasing soil fertility and microbial diversity, the larger aggregate size agglomerate structure provides better physical protection of the organic carbon and its components therein [42], and soil aggregates are closely related to the structure of microbial communities by virtue of their physical properties, their number, and the different sizes [43]. Naturally, variations in the types and contents of carbon and nitrogen sources can lead to changes in soil microbial diversity, and changes in microbial community structure and diversity can also alter soil nutrient cycling and forest ecological functional diversity [44]. In the context of global warming, different forest types and stand structures have significant impacts on understory litter composition, available substrates, and microclimate, resulting in differences in soil microbial diversity. These differences are the result of interactions between environmental factors and forest types.

### 4.2. Soil Aggregate Microbial Community Structure in Different Forest Types

Bacteria constitute a significant and abundant group of soil microorganisms, playing a pivotal role in soil material and energy cycling [45]. This study revealed that the genus composition of soil aggregate bacteria differs between pure *C. hystrix* forests and those mixed with *A*. *crassicarpa*. However, the dominant bacterial phyla in soil aggregates of different forest types and particle sizes were consistent. The dominant bacterial phyla at the phylum level in four different forest types were Proteobacteria, Acidobacteria, Actinobacteria, and Chloroflexi, with their relative abundances accounting for 75.55%–86.21% of the total sequence count. This is similar to the findings of Zhang et al. [46]. Proteobacteria, Acidobacteria, and Actinobacteria have often been found in different forest ecosystem studies [47], indicating their strong adaptability to forest environments [48] and broad ecological niches. Acidobacteria are typically oligotrophic bacteria, while Proteobacteria are eutrophic. In this study, the abundance of Acidobacteria was lowest in mixed *C. hystrix* and *A. melanoxylon* forests (MCA), despite the relatively high soil organic carbon and total nitrogen contents in MCA compared to other forest types. This result aligns with the conclusion of Li et al. [49] that the abundance of this bacterial group is inversely related to nutrient levels in the soil. Actinobacteria are Gram-positive bacteria capable of degrading cellulose and lignin [50]. Their relative abundance in soil aggregates of different particle sizes was much higher in MCA than in the other three forest types, possibly due to the relatively high lignin and cellulose content in MCA, providing a favorable living environment for Actinobacteria. This result is similar to the findings of Gao et al. [39].

Fungi, as important decomposers, collaborate with other microorganisms to decompose soil organics. They can also form symbiotic structures with trees to acquire nutrients and stimulate tree growth, playing a crucial role in soil energy flow and nutrient cycling [51]. Additionally, vegetation type can influence soil fungal community structure [52]. In this study, differences in the genus composition of soil aggregate fungi were observed between mixed forests and pure *C. hystrix* forests. At the phylum level, the dominant fungal communities in soil were Ascomycota and Basidiomycota, but there were differences in their relative abundances among different forest types. Except for mixed *C. hystrix* and *M. macclurei* forests (MCM), the dominant fungal communities in the other forest types accounted for over 90% of the relative abundance of fungi in all samples, which is similar to the findings of Song et al. [53]. Ascomycota are widely distributed in soil ecosystems [54] and have strong adaptability to environmental stress. Saprotrophic fungi within this group can decompose refractory organics in soil, thereby enhancing nutrient utilization efficiency and accelerating soil carbon cycling. In this study, Ascomycota had the highest relative average abundance in pure *C. hystrix* forests (CK), suggesting that pure *C. hystrix* forests may provide an ideal environment for Ascomycota. Basidiomycota, mainly comprising saprotrophic fungi and endomycorrhizal fungi, serve as an important bridge for material exchange between soil and plants [55,56]. We found decreases in Ascomycota abundance and increases in Basidiomycota abundance in all three types of *C. hystrix* mixed forests, indicating that mixing may enhance the mutual cooperation between plants and fungi. This enhancement of cooperative symbiotic modes has also been observed in a study of mixed *P. massoniana* and *Schima superba* forests [57]. Additionally, the relative abundance of Mortierellomycota was also high, with the highest abundance in MCM. This is consistent with the findings of Yuan et al. [58], possibly because Mortierellomycota, as a key group driving soil carbon cycling [59], are more widely distributed in organic-rich and healthy soils.

The environment formed by soil aggregates of different sizes is different, which may cause the utilization strategy of microorganisms to change with the change in environment, for example, soil aggregates change from nutrient-rich to nutrient-poor conditions or from chemoenergetic to chemoenergetic-heterotrophic communities [60] and thus produce microbial community structure with different functions [61,62]. Changes in soil nutrient levels can significantly affect microbial community structure, and different nutrient contents can influence microbial community structure and functional diversity [63]. Interspecific competition among plants, the mixed composition of coniferous and broad-leaved trees, and changes in vegetation diversity can all cause variations in organic composition, total amount and type of plant litter, and its biomass [64], leading to changes in soil nutrient elements and subsequently altering the distribution pattern and structural composition of soil microbial communities. In summary, mixing can alter the composition of soil microbial communities, and soil aggregates at different sizes also influence microbial community structure.

### 4.3. Relationship Between Environmental Factors and Soil Aggregate Microorganisms

The abundance and community composition of soil microbial communities are closely related to environmental factors, interacting with and exhibiting significant correlations among them [65]. The results of this study found that soil aggregate microbial diversity is primarily significantly correlated with NH_4_^+^-N, pH, OC, and C/P, with NH_4_^+^-N being a key factor. Dongqiang et al. [66] found in comparative studies that NH_4_^+^-N plays a crucial role in promoting the development of soil microbial communities, which is similar to the findings of this study. NH_4_^+^-N, as a form of nitrogen that can be directly utilized by soil microorganisms, plays a significant role in activating and promoting microorganisms involved in the nitrogen cycle [67]. It accelerates the recovery of soil microorganisms and enhances the activity of nitrifying bacteria, thereby facilitating soil nitrogen cycling and supporting higher microbial diversity. Soil aggregate pH and AP are key factors influencing the microbial community structure of aggregates in four forest types of *C. hystrix*, which is similar to the findings of Yan Feng et al. [68]. Soil pH is generally considered an important soil variable involved in many soil microbial chemical reaction processes [69], such as affecting enzyme production and activity and cell membrane permeability. It can also alter soil ecosystems by influencing the utilization efficiency of C and N elements, thereby affecting microbial community structure [70].

The microenvironment and uneven distribution of nutrients in different aggregate sizes provide a spatially heterogeneous habitat for microorganisms, which is characterized by differences in resource availability, water content, oxygen concentration, and predation pressure [71], and these differences result in different microbial community structures and diversity. In addition to soil environmental factors, litter decomposition efficiency, root biomass and secretions, and water and heat conditions in related active layers of various vegetation are also important factors influencing the formation of soil microbial community structure. To better understand the ecosystems of pure and mixed forests, future research should further conduct multidimensional experiments to better elucidate the mechanisms of microbial community structure and function of soil aggregates.

## 5. Conclusions

This study demonstrates that establishing mixed forests significantly enhances the diversity and variability of bacterial and fungal communities within soil aggregates, particularly in small-sized aggregates (<0.25 mm), where microbial diversity is more abundant. Compared to pure *C. hystrix* forests (CK), mixed forests exhibit marked differences in the fungal and bacterial composition of soil aggregates. Although both forest types share dominant bacterial phyla such as Proteobacteria, Acidobacteria, Actinobacteria, and Chloroflexi, and dominant genera including *Acidothermus*, *Acidobacterium*, *Solibacter* Candidate, and *Brucella*, the dominant fungal phyla are Basidiomycota, Ascomycota, Mortierellomycota, and Mucoromycota, with dominant genera being uncertain-positioning genus *Thelephoraceae*, *Clavulina*, *Russula*, and *Mortierella*. Among bacterial communities, mixed *C. hystrix* and *A. crassicarpa* plantations’ (MCA) community structure significantly differs from the other three community types. In fungal communities, MCA and mixed *C. hystrix* and *M. laosensis* plantations (MCM) cluster separately, while the community structures of CK and mixed *C. hystrix* and *P. massoniana* plantations (MCP) are relatively similar. Additionally, mixing with *A. crassicarpa* significantly altered the bacterial community structure of soil aggregates, while mixing with both *A. crassicarpa* and *M. laosensis* had a substantial impact on the fungal community structure, resulting in significant differences compared to pure *C. hystrix* forests. Among the factors influencing microbial diversity in soil aggregates, NH_4_^+^-N, pH, and OC play dominant roles, while pH and AP are key environmental factors affecting the bacterial and fungal community structures within soil aggregates. The findings of this study provide theoretical support for a deeper understanding of the mechanisms through which mixed forests influence soil aggregate microbial communities and offer scientific guidance for establishing mixed forests to fully harness their exceptional ecological benefits.

## Figures and Tables

**Figure 1 microorganisms-13-00578-f001:**
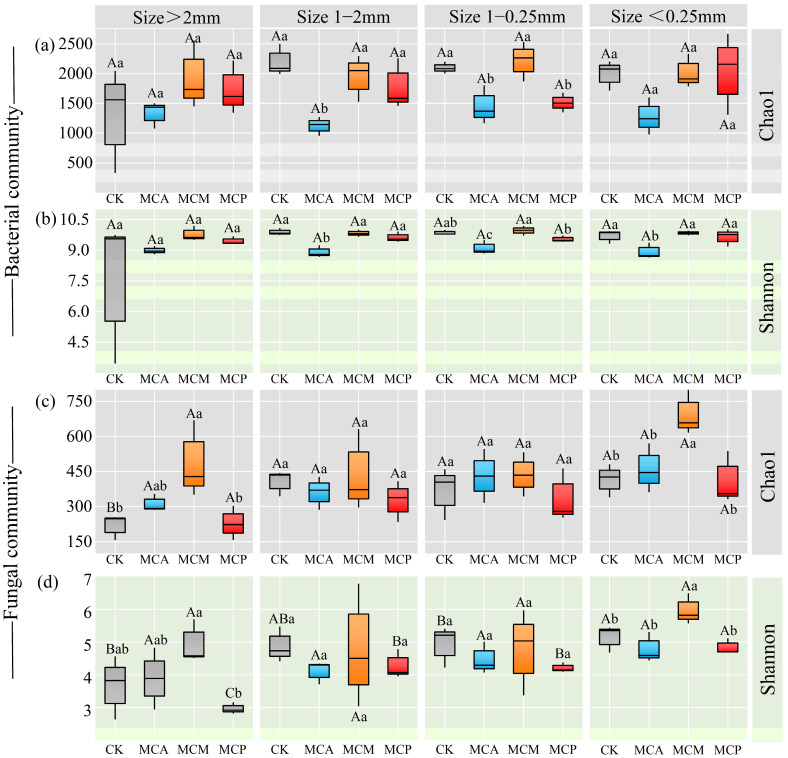
Bacterial (**a**,**b**) and fungal (**c**,**d**) diversity in pure and mixed *C. hystrix* plantations with different aggregate sizes. CK, pure *C. hystrix* plantations; MCA, mixed *C. hystrix* and *A. crassicarpa* plantations; MCM, mixed *C. hystrix* and *M. laosensis* plantations; MCP, mixed *C. hystrix* and *P. massoniana* plantations. Capital letters indicate differences in aggregate sizes of same stand type at the 0.05 significant level; lowercase letters indicate differences in stand types of same aggregate size at the 0.05 significant level. Gray, CK; Blue, MCA; Orange, MCM; Red, MCP.

**Figure 2 microorganisms-13-00578-f002:**
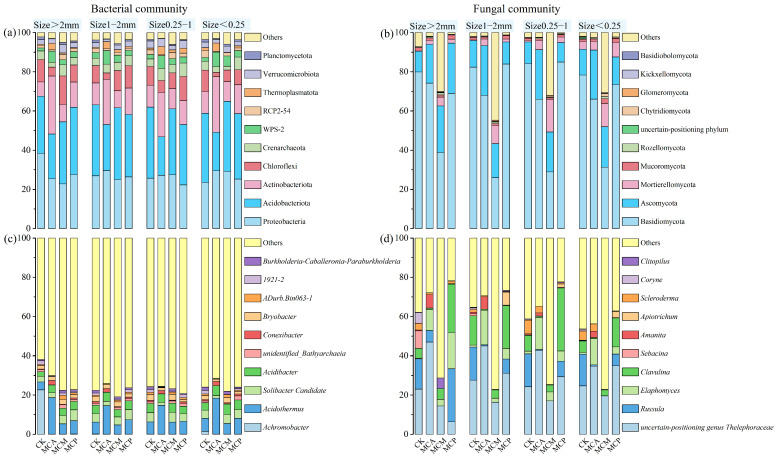
Top ten bacterial phylum (**a**) and genus (**c**), and fungal phylum (**b**) and genus (**d**) in relative abundance in pure and mixed *C. hystrix* plantations with different aggregate sizes. CK, pure *C. hystrix* plantations; MCA, mixed *C. hystrix* and *A. crassicarpa* plantations; MCM, mixed *C. hystrix* and *M. laosensis* plantations; MCP, mixed *C. hystrix* and *P. massoniana* plantations.

**Figure 3 microorganisms-13-00578-f003:**
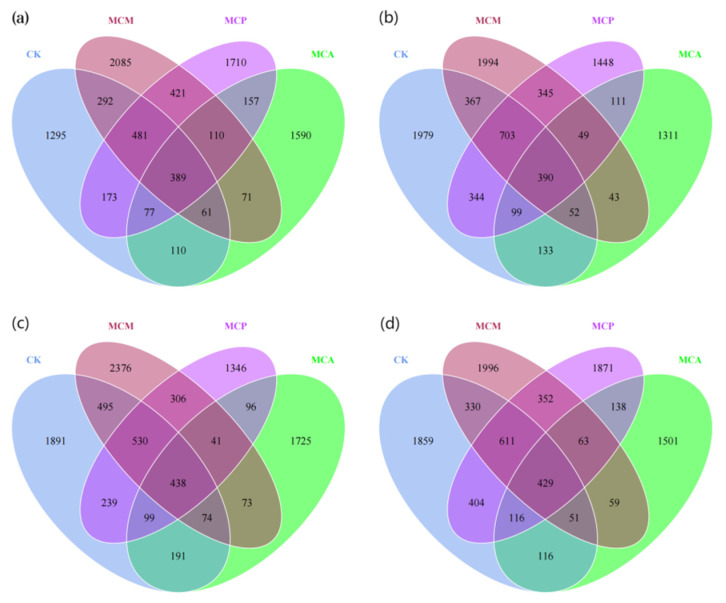
ASV Venn plots of bacterial communities of different aggregate sizes in each stand type. (**a**), size >2 mm; (**b**), size 1–2 mm; (**c**), size 0.25–1 mm; (**d**), size <0.25 mm; CK, pure C. hystrix plantations; MCA, mixed *C. hystrix* and *A. crassicarpa* plantations; MCM, mixed *C. hystrix* and *M. laosensis* plantations; MCP, mixed *C. hystrix* and *P. massoniana* plantations.

**Figure 4 microorganisms-13-00578-f004:**
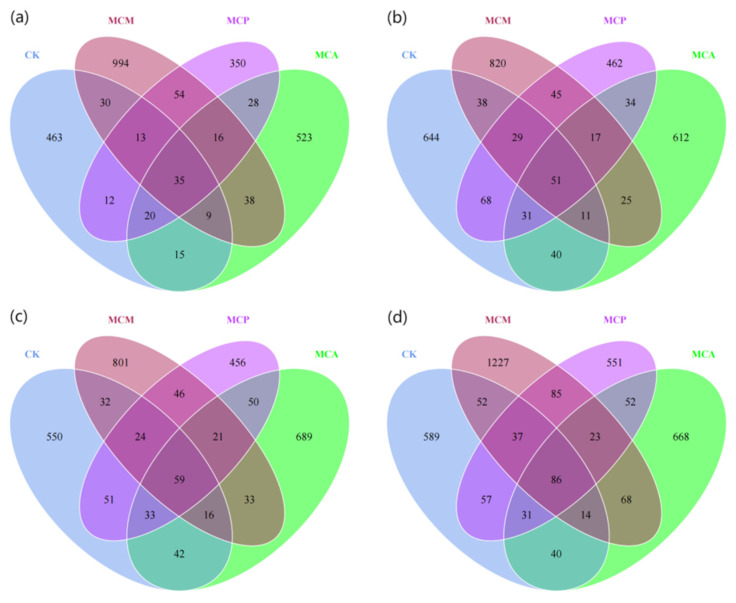
ASV Venn plots of fungal communities of different aggregate sizes in each stand type. (**a**), size >2 mm; (**b**), size 1–2 mm; (**c**), size 0.25–1 mm; (**d**), size <0.25 mm; CK, pure *C. hystrix* plantations; MCA, mixed *C. hystrix* and *A. crassicarpa* plantations; MCM, mixed *C. hystrix* and *M. laosensis* plantations; MCP, mixed *C. hystrix* and *P. massoniana* plantations.

**Figure 5 microorganisms-13-00578-f005:**
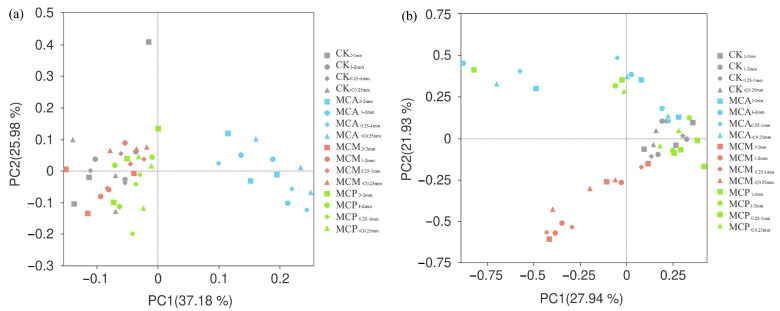
PcoA of bacteria (**a**) and fungi (**b**) with different aggregate sizes in each forest stand. CK, pure *C. hystrix* plantations; MCA, mixed *C. hystrix* and *A. crassicarpa* plantations; MCM, mixed *C. hystrix* and *M. laosensi*s plantations; MCP, mixed *C. hystrix* and *P. massoniana* plantations.

**Figure 6 microorganisms-13-00578-f006:**
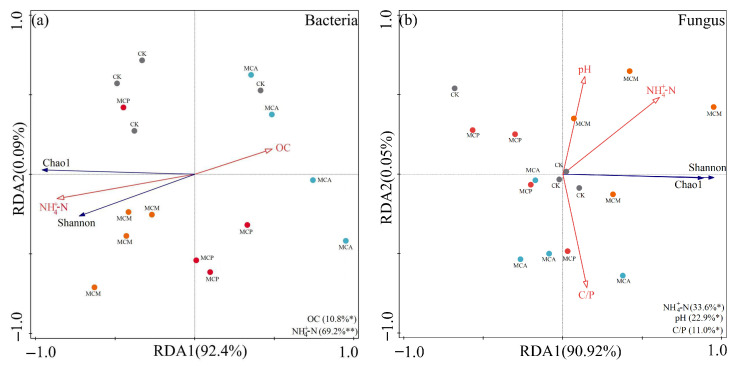
RDA analysis of bacterial (**a**) and fungal (**b**) diversity and chemical factors of soil aggregates in various stand types. Blue arrows indicate microbial diversity indices, red arrows indicate physicochemical factors, circles of each color indicate different stand types. CK, pure *C. hystrix* plantations; MCA, mixed *C. hystrix* and *A. crassicarpa* plantations; MCM, mixed *C. hystrix* and *M. laosensi*s plantations; MCP, mixed *C. hystrix* and *P. massoniana* plantations. OC, organic carbon; NH_4_^+^-N, ammonium nitrogen; C/P, carbon to phosphorus ratio. * and ** denote significance at *p* < 0.05 and *p* < 0.01, respectively.

**Figure 7 microorganisms-13-00578-f007:**
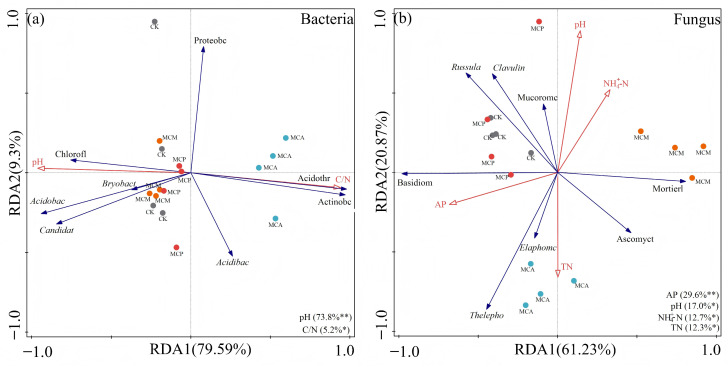
RDA analysis of dominant bacteria (**a**) and fungi (**b**) with soil aggregate chemical factors in various stand types. Blue arrows indicate dominant microorganisms (top four phyla and genera), red arrows indicate physicochemical factors, circles of each color indicate different stand types. CK, pure *C. hystrix* plantations; MCA, mixed *C. hystrix* and *A. crassicarpa* plantations; MCM, mixed *C. hystrix* and *M. laosensis* plantations; MCP, mixed *C. hystrix* and *P. massoniana* plantations. C/N, carbon to nitrogen ratio; AP, available phosphorus; NH_4_^+^-N, ammonium nitrogen; TN, total nitrogen. * and ** denote significance at *p* < 0.05 and *p* < 0.01, respectively.

**Figure 8 microorganisms-13-00578-f008:**
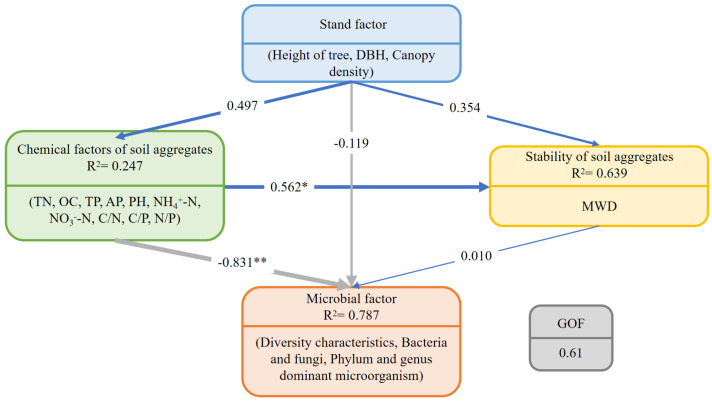
PLS-PM of the effects of forest stand factors, soil aggregate chemistry, and aggregate stability on microbial factors. TN, total nitrogen; OC, organic carbon; TP, total phosphorus; AP, available phosphorus; NH_4_^+^-N, ammonium nitrogen; NO_3_^−^-N, nitrate nitrogen; C/N, carbon to nitrogen ratio; C/P, carbon to phosphorus ratio; N/P, nitrogen to phosphorus ratio; MWD, mean weight diameter; DBH, diameter at breast height. * and ** denote significance at *p* < 0.05 and *p* < 0.01, respectively.

**Table 1 microorganisms-13-00578-t001:** Basic information on sample plots.

Stand Types	Altitudes (Masl)	Slope (°)	CD	DBH (cm)		Tm (m)	
				Main Species	Mixed-Breed Tree Species	Main Species	Mixed-Breed Tree Species
CK	226	27	0.7	22.7	–	18.7	–
MCA	165	25	0.8	20.0	36.0	18.1	25.4
MCM	173	26	0.8	19.8	24.3	21.7	23.5
MCP	182	28	0.7	21.6	23.1	17.1	19.5

Note: CK, pure *C. hystrix* plantations; MCA, mixed *C. hystrix* and *A. crassicarpa* plantations; MCM, mixed *C. hystrix* and *M. laosensis* plantations; MCP, mixed *C. hystrix* and *P. massoniana* plantations. CD, canopy density; DBH, diameter at breast height; TH, tree height.

**Table 3 microorganisms-13-00578-t003:** Physico-chemical properties of whole soil in different stand types.

Indicator	Stand Type			
	CK	MCA	MCM	MCP
BD (g/cm^3^)	1.20 ± 0.04 ab	1.13 ± 0.09 b	1.27 ± 0.03 a	1.16 ± 0.06 ab
SP (%)	49.7 ± 2.65 a	47.5 ± 4.20 a	49.6 ± 3.04 a	50.2 ± 2.49 a
WC (%)	30.1 ± 3.06 a	29.2 ± 4.74 a	32.5 ± 1.76 a	29.3 ± 1.76 a
pH	4.51 ± 0.04 a	4.06 ± 0.06 c	4.48 ± 0.03 ab	4.43 ± 0.04 b
OC (g/kg)	30.6 ± 2.12 ab	40.1 ± 1.49 a	27.7 ± 4.50 ab	21.5 ± 1.50 b
TN (g/kg)	2.16 ± 0.22 b	2.58 ± 0.10 a	1.59 ± 0.14 c	1.94 ± 0.19 b
TP (g/kg)	0.26 ± 0.01 a	0.23 ± 0.02 ab	0.25 ± 0.02 ab	0.22 ± 0.03 b
AP (mg/kg)	11.3 ± 0.15 a	8.42 ± 1.10 b	8.03 ± 1.17 b	8.10 ± 0.84 b
NH_4_^+^-N (mg/kg)	75.0 ± 8.01 a	34.9 ± 10.6 b	70.4 ± 11.4 a	47.7 ± 7.08 b
NO_3_^−^-N (mg/kg)	3.22 ± 0.14 a	3.60 ± 0.86 a	4.29 ± 0.83 a	3.88 ± 0.16 a
C/N	14.2 ± 0.53 ab	15.6 ± 0.01 a	17.6 ± 4.05 a	11.1 ± 1.38 b
N/P	8.17 ± 0.38 bc	11.1 ± 1.19 a	6.30 ± 0.73 c	8.98 ± 1.54 b
C/P	116 ± 2.97 b	174 ± 18.5 a	109 ± 13.3 b	98.6 ± 5.49 b

Note: CK, pure *C. hystrix* plantations; MCA, mixed *C. hystrix* and *A. crassicarpa* plantations; MCM, mixed *C. hystrix* and *M. laosensis* plantations; MCP, mixed *C. hystrix* and *P. massoniana* plantations. BD, bulk density; SP, soil porosity; WC, water content; OC, organic carbon; TN, total nitrogen; TP, total phosphorus; AP, available phosphorus; NH_4_^+^-N, ammonium nitrogen; NO_3_^−^-N, nitrate nitrogen; C/N, carbon to nitrogen ratio; C/P, carbon to phosphorus ratio; N/P, nitrogen to phosphorus ratio. Lowercase letters indicate differences in stand types of same aggregate size at the 0.05 significant level.

## Data Availability

The raw data supporting the conclusions of this article will be made available by the authors on request.

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
