# Peer review of "Impact of Tree Species Mixture on Microbial Diversity and Community Structure in Soil Aggregates of *Castanopsis hystrix* Plantations"

_microorganisms, 2025, doi:10.3390/microorganisms13030578_

Round 1
Reviewer 1 Report
Comments and Suggestions for Authors
Introduction
Line 75. Please add “(Pinopsida:Cupressaceae)” after the “Castanopsis hystrix”
Line 80. Please add “organic acids” before “and others…”
Line 100. Please add “(Magnoliopsida:Fabaceae)” after “Acacia crassicarpa”
Line 101 Please add “(Pinopsida:Pinaceae)” after “Pinus massoniana”
Line 101 Please add “(Magnoliopsida: Hamamelidaceae)” after “Mytilaria laosensis”
Line 107. Please elaborate the presented hypothesis of the expected impact of mixed species at ecological (interactions), physiological (nutrient cycling, plant exhudates) and microbial-association levels (microbiota architecture shifts) on soil aggregation.
Materials and methods
Line 113. Please add the province and “China” after “Park”
Line 115. Please substitute “m” with “masl”. Please remove “an annual accumulated temperature of around 7500°C”
Line 128. Please change “3m×4m” with “3 × 4 m”
Line 130. Please specify the dosage and formulation of the applied fertilizer. Please clarify whether weeds were mechanically removed or an herbicide was applied, including formula and dosage.
Line 131. Please briefly describe how slope, canopy density, DBH, and tree height were calculated. Please state if those measures correspond to experimental plots or a general area.
Line 134. Table 1. Please substitute “m” with “masl” in “Altitudes” heading. If these determinations are part of your results (not published previously. Please move to the results section.
Line 140. Please substitute “20m×20m” with “20 ×20 m”
Line 141. Please add an space after “25”
Line 148. Please substitute “Under relatively” with “Providing the best possible”. Please add examples of these conditions (wearing gloves, using sterilized utensils for sample manipulation, etc.)
Line 158. Please add “amplicon” after “sequencing”
Line 179-181. Please move “The physicochemical…”, Table 2, and Table 3 to the results section.
Line 182. Table 2. Please add a space before abbreviations and units in the Indicator column as “BD (g/cm3)” and “SP (%)”, etc. Please justify data in the rest of the columns, referencing the decimal point. Please add a space after and before the “±” symbol in all cases. Please add a statistical test using indicator and stand type.
Line 189. Table 3. Please justify data in the rest of the columns, taking the decimal point as a reference. Please add a space after and before the “±” symbol in all cases. Please add a statistical test using indicators, stand type, and soil aggregate size.
Line 197. Please cite the reactants, commercial software, and materials as “name (catalog number, manufacturer, city, and country of manufacturer’s headquarters)”
Line 199. Please add the primers’ reference.
Line 208. Please add a space before “Quality”
Line 209. Please cite Trimmomatic.
Line 210. Please cite cutadapt. Please add a space before “Paired”
Line 213. Please cite cutadapt. Please add a space before “Chimera”. Please cite UCHIME.
Line 214. Please add a space before “Denoising”. Please cite DADA2.
Line 217. Please add “Microsoft” before “Excel”
Line 219. Please cite SPSS as commercial software.
Line 225. Please cite Origin 2022.
Line 229. Please cite Canoco and R software.
Line 230. Please cite the plspm package.
Results
Line 232. Please add a new section for soil analysis. Present here the results shown in Tables 2 and 3.
Line 233. Please mention the precise number of sequenced samples, the total amount of bp obtained, and the average per sample. Please add the total number of ASV obtained in all samples, mentioning the global percentage of those sequences that were not taxonomically assigned.
Lines 238-240. Please move “This indicates… …aggregates” to the discussion section.
Lines 245-246. Please move “This suggests… …aggregates” to the discussion section.
Lines 249-253. Please move this paragraph below the figure as a footnote.
Lines 264-266. Please move “This suggests… …stands” to the discussion section.
Lines 272, 273. Please substitute “Candidatus_Solibacter” with “Solibacter Candidate”
Lines 275-26. Please move “This indicates… …stands” to the discussion section.
Lines 285-287. Please move “This suggests… …stands” to the discussion section.
Lines 289, 291, 293, 295. Please substitute “Thelephoraceae_gen_Incertae_sedis” with “uncertain-positioned genus Thelephoraceae”
Lines 297-298. Please move “This demonstrates… … stands” to the discussion section.
Line 300. Figure 2 Please italicize Latin names (genera). Please substitute “Fungi_phy_Incertae_sedis” with “uncertain-positioned phylum”. Please substitute “Thelephoraceae_gen_Incertae_sedis” with “uncertain-positioned genus Thelephoraceae”. Please use different colors for fungal taxons, as blue variants are too similar, making the figure hard to read. Please remake the figure to show the bacteria panels on the left and the fungus on the right.
Line 321. Figure 3. Please use the same color code for each treatment as in the Figure 1. Please remove from main text and present it as supplementary material.
Line 329. Figure 4. Please use the same color code for each treatment as in Figure 1. Please remove it from the main text and present it as supplementary material.
Line 351. Please add a PERMANOVA analysis for the hypothesis test on the microbial structure between stands and soil aggregation.
Line 352. Figure 5. Please use the same color code for each treatment as in Figure 1. Please move the legend to the figure footnote
Line 372. Figure 6. Please use the same color code for each treatment as in Figure 1.
Line 370. Please remove Table 4 from the main text and present it as supplementary material. Instead please include a PERMANOVA analysis for hypothesis test of physicochemical factors and microbial diversity by stand and by soil aggregation.
Line 396. Please substitute “demonstrate” with “show”
Line 401. Please substitute “Thelephoraceae_gen_Incertae_sedis” with “uncertain-positioned genus Thelephoraceae”.
Line 406. Please use the same color code for each treatment as in Figure 1.
Line 370. Please remove Table 5 from the main text and present it as supplementary material. Instead, please include a PERMANOVA analysis for hypothesis tests of physicochemical factors and bacterial or fungal diversity.
Discussion
Please incorporate here the ideas relocated from the results section
Conclusions
Line 579. Please substitute “Candidatus_Solibacter” with “Solibacter Candidate”
Line 581. Please substitute “Thelephoraceae_gen_Incertae_sedis” with “uncertain-positioned genus Thelephoraceae”.
Reviewer 2 Report
Comments and Suggestions for Authors
The manuscript, titled "Impact of Tree Species Mixture on Microbial Diversity and Community Structure in Soil Aggregates of Castanopsis hystrix Plantations," examines a soil factor important for artifically planted forest communities. It makes findings regarding the relationship between the species composition of the dominant tree species and the microbial communities formed on soil particles.
The introduction is adequate and well detailed. The presentation of the materials, methods and statistical procedures used is also sufficiently detailed.
The presentation of the results is sufficiently detailed and easy to understand.
In the case of Figure 1, the explanation of each color is missing, and the marking of the 4 forest types is incompletely visible at the bottom of the figure. Please correct this!
The conclusions are appropriate and comparative. A sufficient number of similar scientific works are cited.
After implementing the clarifications and modifications suggested above, I recommend publishing the manuscript as a scientific article.
